# Girdin is a component of the lateral polarity protein network restricting cell dissemination

**Cornélia Biehler**[1,2], **Li-Ting Wang**[3], **Myriam Sévigny**[1,2], **Alexandra Jetté**[1,2], **Clémence L. Gamblin**[1,2], **Rachel Catterall**[3], **Elise Houssin**[1,2¤], **Luke McCaffrey**[3,4], **Patrick Laprise**[1,2]*

**1** Centre de Recherche sur le Cancer, Université Laval, Québec, Canada, **2** axe oncologie du Centre de Recherche du Centre Hospitalier, Universitaire de Québec-UL, Québec, Canada, **3** Rosalind and Morris Goodman Cancer Research Centre, McGill University, Montreal, Canada, **4** Gerald Bronfman Department of Oncology, McGill University, Montreal, Canada

¤ Current address: Univ Rennes, CNRS, IGDR (Institut de Génétique et Développement de Rennes), Rennes, France

* Patrick.Laprise@crchudequebec.ulaval.ca

**Data Availability Statement:** All relevant data are within the manuscript and its Supporting Information files.

## Abstract

Epithelial cell polarity defects support cancer progression. It is thus crucial to decipher the functional interactions within the polarity protein network. Here we show that *Drosophila* Girdin and its human ortholog (GIRDIN) sustain the function of crucial lateral polarity proteins by inhibiting the apical kinase aPKC. Loss of GIRDIN expression is also associated with overgrowth of disorganized cell cysts. Moreover, we observed cell dissemination from *GIRDIN* knockdown cysts and tumorspheres, thereby showing that GIRDIN supports the cohesion of multicellular epithelial structures. Consistent with these observations, alteration of *GIRDIN* expression is associated with poor overall survival in subtypes of breast and lung cancers. Overall, we discovered a core mechanism contributing to epithelial cell polarization from flies to humans. Our data also indicate that GIRDIN has the potential to impair the progression of epithelial cancers by preserving cell polarity and restricting cell dissemination.

## Author summary

Epithelia, composed of epithelial cells, delimit the frontier between the external environment and the inside of complex organisms. Therefore, epithelial cells cover the surface of the body (e.g. skin) and line internal cavities of organs (found in the intestine, liver, lungs, etc). An important function of epithelia is to selectively transport specific molecules to adjust the chemical composition of the different body compartments. This function relies on the asymmetric distribution of many cellular constituents, a structural organization referred to as epithelial polarity. The polarized architecture of epithelial cells is also required to maintain tissue homeostasis, as loss of epithelial polarity contributes to cancer progression. Here, we show that the protein GIRDIN is essential to maintain epithelial polarity in fruit flies and human cells. In addition, the absence of GIRDIN causes cell dissemination from tumor-like structures. This process is reminiscent to the formation of metastases (secondary tumors), which are the primary cause of mortality in cancer

**Funding:** This work was supported by operating grants from the Canadian Institute of Health Research (CIHR) to P.L. (MOP-142236) and L.M. (PJT-156271). P.L. and L.M. are Fonds de la Recherche du Quebec – Santé Research Scholars. The work was also supported by studentships from the Centre de Recherche sur le Cancer de l'université Laval (C.B.), Karassik Family Foundation (L.T-W) and Defi Canderel (R.C.). The funders had no role in study design, data collection and analysis, decision to publish, or preparation of the manuscript.

**Competing interests:** The authors have declared that no competing interests exist.

patients. It is thus not surprising that our data indicate that low GIRDIN levels are associated with a poor prognosis in some cancers. Overall, our study identifies GIRDIN as a potential target in cancer.

## Introduction

The ability of epithelia to form physical barriers is provided by specialized cell-cell junctions, including the *zonula adherens* (ZA). The latter is a belt-like adherens junction composed primarily of the transmembrane homotypic receptor E-cadherin, which is linked indirectly to circumferential F-actin bundles through adaptor proteins such as β-catenin and α-catenin [1,2]. In *Drosophila* embryonic epithelia, the protein Girdin stabilizes the ZA by reinforcing the association of the cadherin–catenin complex with the actin cytoskeleton [3]. This function in cell–cell adhesion is preserved in mammals, and supports collective cell migration [4,5]. Fly and human Girdin also contribute to the coordinated movement of epithelial cells through the organization of supracellular actin cables [3,4].

In addition to creating barriers, epithelial tissues generate vectorial transport and spatially oriented secretion. The unidirectional nature of these functions requires the polarization of epithelial cells along the apical-basal axis. In *Drosophila*, the scaffold protein Bazooka (Baz) is crucial to the early steps of epithelial cell polarization, and for proper assembly of the ZA [6–9]. Baz recruits atypical Protein Kinase C (aPKC) together with its regulator Partitioning defective protein 6 (Par-6) to the apical membrane [10–12]. The small GTPase Cdc42 contributes to the activation of aPKC and p21-activated kinase (Pak1), thereby acting as a key regulator of cell polarity [13–16]. Baz also contributes to apical positioning of the Crumbs (Crb) complex, which is composed mainly of Crb, Stardust (Sdt), and PALS1-associated Tight Junction protein (Patj) [10,17]. Once properly localized, the aPKC–Par-6 and Crb complexes promote the apical exclusion of Baz, which is then restricted to the ZA [12,18,19]. The apical exclusion of Baz is essential to the positioning of the ZA along the apical-basal axis [18], and for full aPKC activation [20–22].

The function of aPKC is evolutionarily preserved, and human atypical PKCι (PKCλ in other mammals) and PKCζ contribute to epithelial cell polarization [23]. aPKC maintains the identity of the apical domain through phospho-dependent exclusion of lateral polarity proteins such as Yurt (Yrt) and Lethal (2) giant larvae (Lgl) [15,24–26]. In return, these proteins antagonize the Crb- and aPKC-containing apical machinery to prevent the spread of apical characteristics to the lateral domain [8,14,15,24,27–31]. In combination with the function of Baz, these feedback mechanisms provide a fine-tuning of aPKC activity in addition to specifying its subcellular localization. This is crucial, as both over- and under-activation of aPKC is deleterious to epithelial polarity in fly and mammalian cells, and ectopic activation of aPKC can lead to cell transformation [11,32–35].

Cell culture work has established that mammalian GIRDIN interacts physically with PAR3 –the ortholog of Baz–and PKCλ [36–38]. Depletion of *GIRDIN* in Madin-Darby Canine Kidney (MDCK) epithelial cells delays the formation of tight junctions in Ca$^{2+}$ switch experiments [38]. GIRDIN is also an effector of AMP-activated protein kinase (AMPK) in the maintenance of tight junction integrity under energetic stress [39]. Moreover, mammalian GIRDIN is required for the formation of epithelial cell cysts with a single lumen, supporting a role for this protein in epithelial morphogenesis as reported in flies [3,36,38,39]. As cyst morphogenesis is linked to epithelial cell polarity [40], these studies suggest that GIRDIN is involved in establishing the apical-basal axis. However, further studies are required to clarify the role of GIRDIN in

apical-basal polarity *per se*, as other cellular processes could explain the phenotype associated with altered GIRDIN expression. For instance, spindle orientation defects impair the formation of epithelial cysts [41]. Of note, PAR3, aPKC, and AMPK are all required for proper spindle positioning in dividing epithelial cells [42–45]. The molecular mechanisms sustaining the putative role of GIRDIN in epithelial cell polarity also need to be better deciphered. Here, we further investigated the role of fly and human Girdin proteins in the regulation of epithelial cell polarity, and showed that these proteins are part of the lateral polarity protein network. One crucial function of Girdin proteins is to repress aPKC function. We also discovered that loss of Girdin proteins promotes overgrowth of cell cysts, and cell dissemination from these multicellular structures. Consistent with these data, we found that low *GIRDIN* expression correlates with poor overall survival in subtypes of breast and lung cancers.

## Results

### *Girdin* is a crucial component of the genetic network supporting lateral membrane stability in polarized epithelial cells

To explore the role of Girdin in epithelial cell polarity regulation, we investigated its functional relationship with Yrt and Lgl, which are known polarity regulators in *Drosophila* embryos. These lateral proteins prevent overactivation of the Crb- and aPKC-containing apical machinery, thereby precluding apicalization of the lateral membrane [8,24,27,30,31]. Similar to Girdin, Yrt and Lgl are provided maternally [3,27,46]. Although suboptimal, the maternal contribution is sufficient to maintain apical-basal polarity in zygotic mutant embryos carrying null alleles for *Girdin*, *yrt* or *lgl* [3,27,46] (Figs 1A-I and 2A–2C). Zygotic loss of these genes thus represents a sensitized background. In contrast, a complete loss of Lgl expression in *lgl* maternal and zygotic (M/Z) mutants is associated with strong polarity defects characterized by ectopic localization of the apical protein Crb to the lateral membrane in post-gastrulating embryos [30]. This phenotype is clearly visible at stage 11 of embryogenesis as shown by the partial co-localization of Crb with the lateral protein Discs large 1 (Dlg1; Fig 1J). At stage 13, the monolayered architecture of the differentiating epidermis is compromised, and polarity defects were attenuated but still apparent (Fig 1K). Toward the end of embryogenesis (stage 16), Lgl-deficient epidermal cells formed polarized cysts of cells with their extended–cuticle secreting–apical membrane facing out (Fig 1L). As a consequence, *lgl* M/Z mutant embryos assembled a highly convoluted cuticle (Fig 1R), whereas zygotic *lgl* and *Girdin* mutant embryos displayed a normal cuticle (Fig 1P and 1Q). Strikingly, a combination of zygotic *lgl* and *Girdin* mutations phenocopied a total loss of Lgl (Fig 1M–1O and 1S). Similarly, depletion of Girdin, which is not sufficient to cause obvious polarity defects [3] (Fig 2D–2F), strongly enhanced the zygotic *yrt* mutant phenotype. Indeed, *Girdin yrt* double mutant specimens were indistinguishable from *yrt* M/Z embryos [27] (Fig 2G–2L, 2O and 2P), and presented an expansion of Crb expression territories at stages 13 of embryogenesis (Fig 2H and 2K). These strong genetic interactions suggest that Girdin cooperates with Yrt and Lgl to maintain lateral characteristics in polarized epithelial cells.

Girdin plays an important role in strengthening DE-cadherin (DE-cad)-dependent cell–cell adhesion in developing embryos [3]. We thus wondered whether alteration of ZA properties could explain the genetic interaction between *Girdin* and *lgl*. We investigated this possibility by combining zygotic loss of Lgl expression with mutant alleles of *shotgun* (*shg*) and *α-Catenin* (*α-Cat*), which encode core ZA components (DE-cad and α-Cat, respectively; [47–49]). The cuticles of *lgl shg* and *lgl* α-*Cat* double mutant embryos were similar to single *shg* or α-*Cat* mutants, respectively (S1A–S1D Fig). Moreover, immunostaining of Crb and Dlg1 revealed no major polarity defects in double mutant embryos (S1H–S1J and S1N–S1P Fig), which were

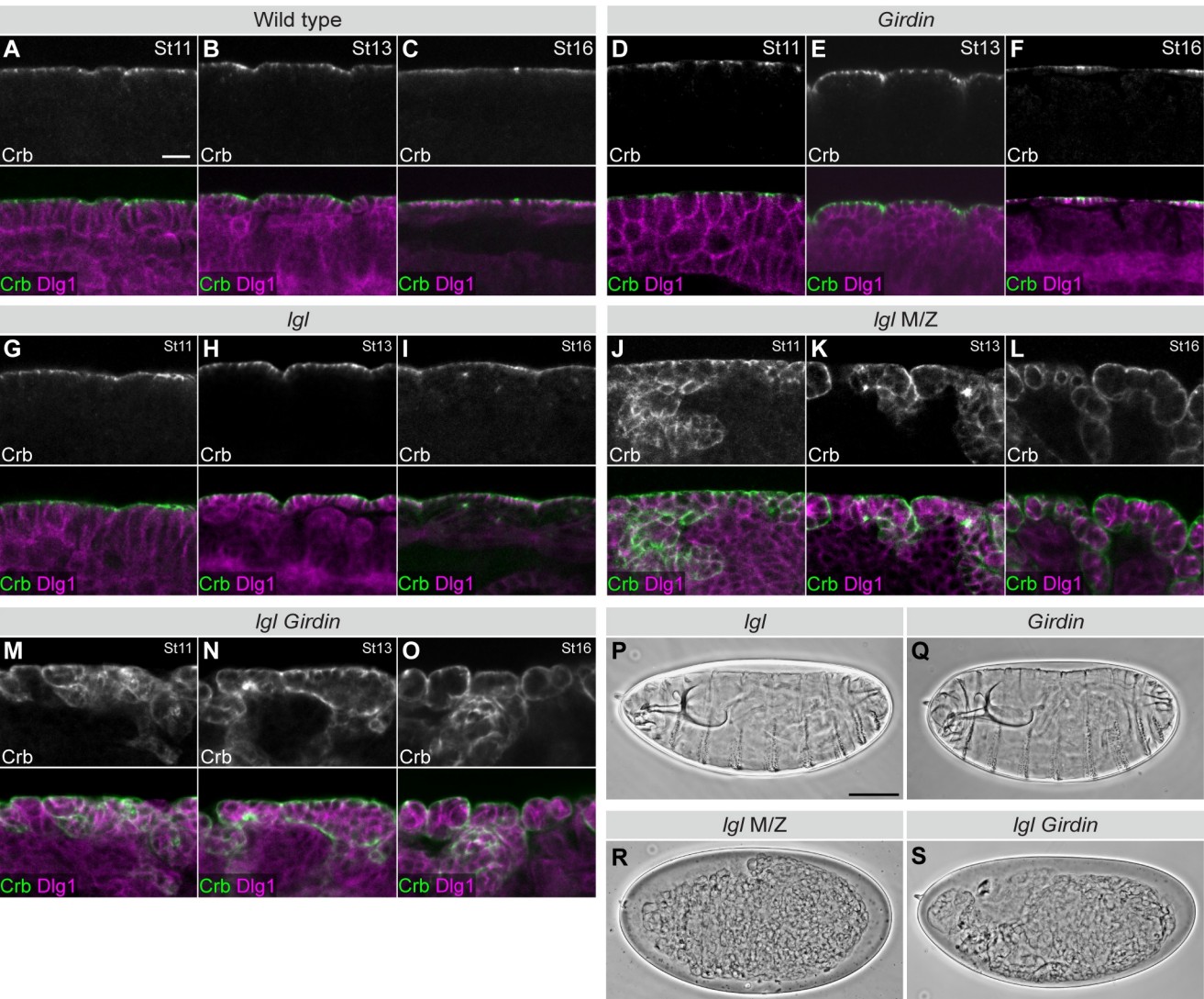

**Fig 1. Girdin cooperates with Lgl to support epithelial cell polarity. A-O**, Embryos of the indicated genotypes were fixed and co-stained for Crb and Dlg1. M/Z stands for maternal and zygotic mutant. Panels depict a portion of the ventral ectoderm or epidermis at stage (St) 11 (**A, D, G, J, M**), stage 13 (**B, E, H, K, N**), or stage 16 (**C, F, I, L, O**) of embryogenesis. In each case, the apical membrane is facing up. Scale bar in **A** represents 10 μm, and also applies to **B-O**. **P-S**, Panels depict cuticle preparations of whole mounted embryos of the indicated genotypes. The anterior part of the embryo is oriented to the left, and the dorsal side is facing up. Scale bar in **P** represents 100 μm, and also applies to **Q-S**. Panels in A-S show representative results taken from experiments that were repeated at least three times [replicate (r) ≥3], and a minimum of 25 embryos of the indicated genotypes were analyzed in each replicate.

similar to single *shg* or *α-Cat* mutant specimens (S1E–S1G and S1K–S1M Fig). This led us to conclude that the weakening of the ZA is not sufficient to enhance the *lgl* mutant phenotype. Although these results do not exclude the possibility that Girdin sustains epithelial cell polarity in part by strengthening cell-cell adhesion, they strongly argue that Girdin has additional roles directly impacting on the protein network supporting the stability of the lateral domain.

## Fly Girdin and human GIRDIN repress the function of aPKC to support epithelial cell polarity

Although Yrt and Lgl control Crb activity independently and in separate time windows during fly embryogenesis [27,28,30], they are both inhibited by aPKC-dependent phosphorylation

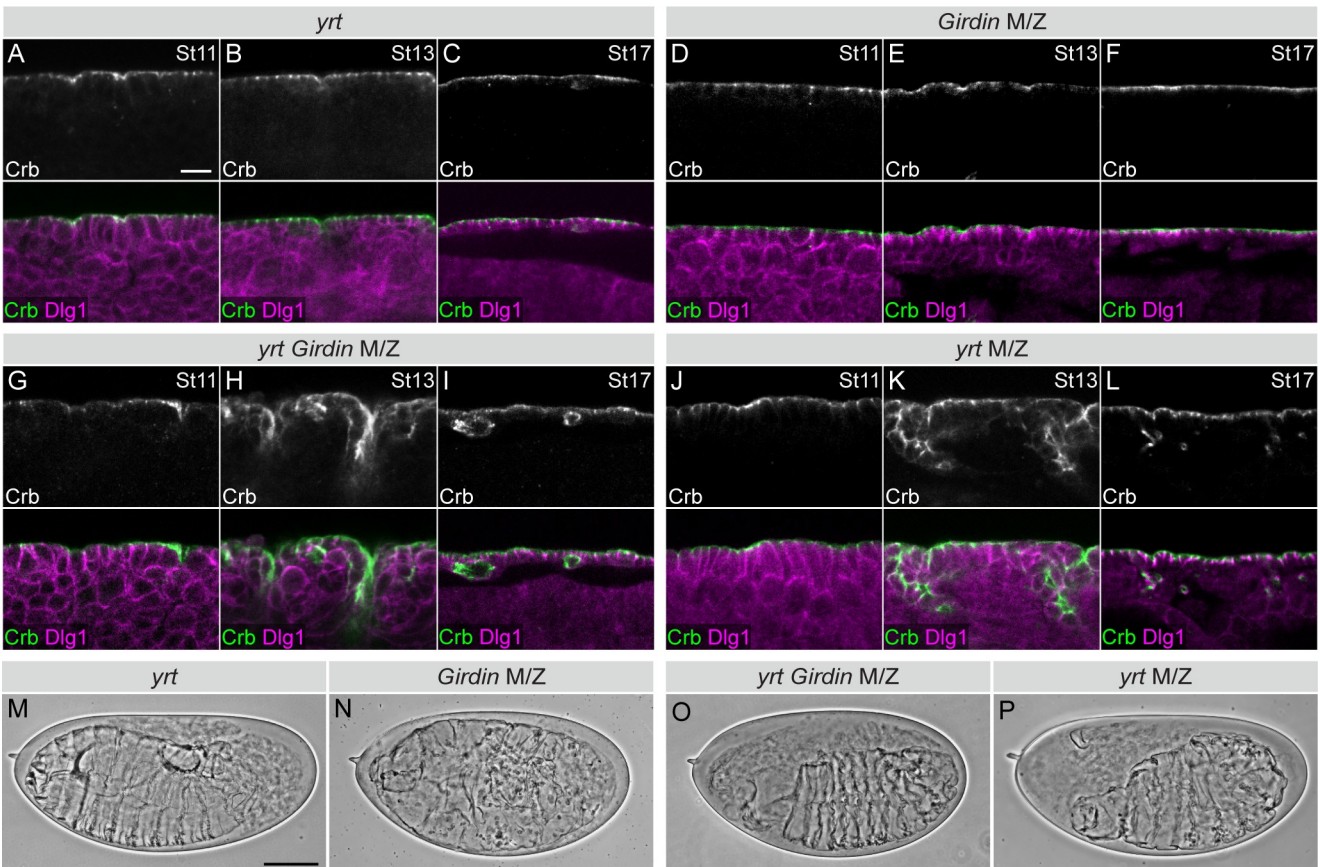

**Fig 2. *Girdin* mutation enhances the delocalization of the apical protein Crb in *yrt* mutants. A-L**, Embryos were immunostained for Crb and Dlg1, and a part of the ventral ectoderm or epidermis was imaged by confocal microscopy (the apical side of the epithelial tissue is facing up). The developmental stage of embryos (St) is indicated in the upper right corner of panels. Scale bar in **A** represents 10 μm, and also applies to **B**-**L**. **M**-**P**, Cuticle preparations of whole mounted embryos are shown. Scale bar in **M** represents 100 μm, and also applies to **N**-**P**. Panels in A-P depict representative results taken from experiments that were repeated at least three times (r ≥3), and a minimum of 55 embryos of the indicated genotypes were analyzed in each replicate.

[24,25,50,51]. It is thus plausible that Girdin supports the function of these parallel pathways by acting as a repressor of aPKC activity, which impacts on the function of both Lgl and Yrt. Accordingly, Yrt, which is a substrate of aPKC [24], showed reduced electrophoretic mobility in *Girdin* null embryos compared to control animals. This was due to enhanced phosphorylation of Yrt, as treatment of samples with the λ Protein Phosphatase abolished the delayed migration profile of Yrt in *Girdin* null embryos (Fig 3A). The aPKC inhibitor CRT-006-68-54 largely suppressed the hyper-phosphorylation of Yrt in *Girdin* null embryos (Fig 3B). These results strongly argue that the activity of aPKC increases in the absence of Girdin. To obtain functional evidence that Girdin represses aPKC activity, we performed genetic interactions. Maternal knockdown of *aPKC* resulted in strong epithelial morphogenesis defects, as revealed by analysis of the cuticle, and a fully penetrant lethality. Indeed, most knocked-down embryos displayed epithelial tissue collapse (referred to as class I embryos, Fig 3C and 3F). Remaining embryos showed a weaker phenotype, and secreted either small patches of cuticle (class II; Fig 3D and 3F), or large continuous sheets of cuticle with differentiated structure such as denticle belts (class III; Fig 3E and 3F). Knockdown of *aPKC* in zygotic *Girdin* mutants, which show a normal cuticle phenotype ([3] and Fig 1Q), significantly changed the distribution of embryos in each category. However, reduction of Girdin levels in *aPKC* knockdown embryos was not sufficient to modify the degree of lethality. Specifically, a decrease in class I embryos

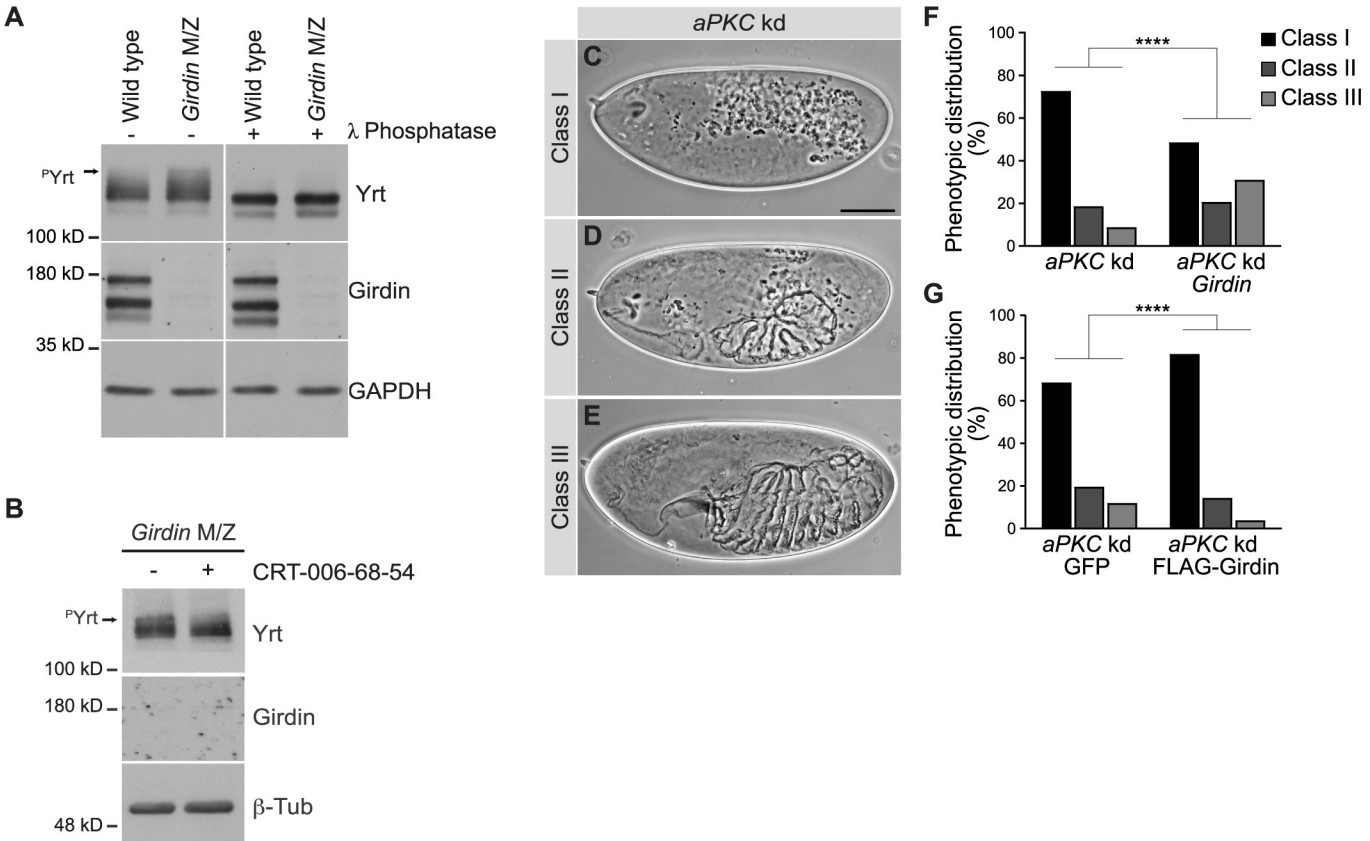

**Fig 3. Girdin antagonizes aPKC functions. A**, Stage 11–13 wild type embryos or maternal and zygotic (M/Z) *Girdin* mutant embryos were homogenized and processed for Western blotting. Where indicated, samples were treated with the λ Protein Phosphatase prior to electrophoresis. Arrow indicates the position of phosphorylated Yrt proteins (PYrt). Immunoblotting of Girdin validates the genotype of embryos, whereas Gapdh1 was used as loading control. **B**, *Girdin* null embryos were incubated in a saline solution in the absence or presence of the aPKC inhibitor CRT-006-68-54 for 1h. Embryos were then homogenized and processed for Western blotting. Arrow points to phosphorylated Yrt proteins (PYrt). Immunoblotting of Girdin confirms the genotype of embryos, and β-Tubulin was used as loading control. **C-E**, Cuticle preparations of *aPKC* knockdown (kd) embryos. Embryos were separated in three classes (I, II, or III) to account for phenotypic variability. Scale bar in **C** represents 100 μm, and also applies to **D** and **E**. **F**, Histogram showing the phenotypic distribution in percentage (%; according to the classes defined in **C-E**) in collections of embryos of the following genotypes: *aPKC* knockdown in a wild type background (*aPKC* kd; *n* = 1312), *aPKC* knockdown in a *Girdin* mutant background (*aPKC* kd *Girdin; n* = 1573). The specified number of embryos (n) represents the sum of specimens analyzed in three independent experiments. **G**, Phenotypic distribution of *aPKC* knockdown embryos expressing GFP (*aPKC* kd GFP; *n* = 603) or FLAG-Girdin (*aPKC* kd FLAG-Girdin; *n* = 697). For **F** and **G**, embryos were analyzed in three blinded experiments, and a Chi-squared test of independence was performed (****: ρ < 0.0001).

concomitant with an increase in class II and III specimens was observed (Fig 3F). This shows that reduction of Girdin levels alleviates the impact of *aPKC* knockdown, and suggests that Girdin normally limits the function of residual aPKC in knocked-down embryos (S2A Fig). In accordance with this conclusion, overexpression of FLAG-Girdin enhanced the severity of the phenotype associated with the knockdown of *aPKC* (Fig 3G). FLAG-Girdin displayed a similar distribution to endogenous Girdin [3], and overexpression of this protein in a wild type background had no impact on epithelial tissue morphogenesis (S2C–S2E Fig). Thus, *Drosophila* Girdin opposes aPKC function, and supports apical-basal polarity in epithelial cells.

To explore the intriguing possibility that human GIRDIN also restricts aPKC activity to control polarity, we used the human colon carcinoma Caco-2 cell line. These cells adopt a monolayered organization in 3D culture, and form hollow cysts with a single lumen. Cells forming the cysts are polarized along the apical-basal axis, with the PAR6- and F-ACTIN-enriched apical domain facing the central lumen (Fig 4A, 4H, 4L and 4N; [41]). PAR6

segregates from the lateral adhesion protein E-CADHERIN (E-CAD; Fig 4H–4L). Consistent with our hypothesis, knockdown of *GIRDIN* in Caco-2 cells mimicked the phenotype associated with increased PKCζ activity (expression of the constitutively active PKCζ–T410E), and resulted in the formation of solid cysts, or cysts with multiple atrophic lumens (Fig 4A, 4B, 4E, 4F and S2B Fig). *GIRDIN* knockdown cells forming the disorganized 3D structures were mispolarized, as shown by the peripheral accumulation of PAR6 (Fig 4I and 4M, arrow) and F-ACTIN (Fig 4O). Despite that *GIRDIN* knockdown cysts displayed a similar size to controls, they contained more cells resulting from lumen filling (Fig 4N–4P). The overgrowth phenotype was exacerbated and resulted in larger cysts when *GIRDIN* knockdown was combined with overexpression of wild type PKCζ, whereas overexpression of the latter had no impact on cyst size (Fig 4C, 4D and 4G). Again, this is consistent with a stronger PKCζ activation in *GIRDIN* knocked-down cells compared to cells expressing a scrambled shRNA, as expression of PKCζ–T410E also caused enlargement of cysts (Fig 4E and 4G). To confirm that aPKC activation contributes to the phenotypes associated with depletion of GIRDIN, we used the aPKC inhibitor CRT-006-68-54. Strikingly, inhibition of aPKC activity suppressed the defects caused by knockdown of *GIRDIN* (Fig 4Q–4U). Expression of the dominant negative aPKC-K281W in *GIRDIN* knockdown cysts resulted in a similar rescue of lumen formation (S3A–S3G Fig). Together, these results indicate that, similar to fly Girdin, GIRDIN restrains aPKC function to support epithelial morphogenesis and apical-basal polarity.

## GIRDIN maintains the cohesion of epithelial cysts

In addition to the morphogenesis and polarity defects associated with decreased GIRDIN expression, we observed the presence of isolated cells or small cell aggregates in the proximity of most *GIRDIN* knocked-down Caco-2 cell cysts (Fig 4B and 4D, arrows). As we previously reported that cell cysts detach from the epidermis and survive outside of it in *Girdin* mutant *Drosophila* embryos [3], we hypothesized that some cells separate from *GIRDIN* knockdown Caco-2 cell cysts. To test this hypothesis, we performed time-lapse microscopy of control (expressing a scrambled shRNA) and *GIRDIN* knocked-down cysts. Over a period of 26 hours, control cysts showed cell–cell rearrangements, which were resolved to maintain the monolayered organization (Fig 5A and S1 Video). In *GIRDIN* knockdown cysts, cells were frequently observed extending from the periphery of cysts and detaching (Fig 5B arrow, 5E–5G and S2 Video). In control cysts, the extensions were less frequent, and detachment was not observed (Fig 5A, 5E–5G and S1 Video). Further analysis also revealed that loss of GIRDIN expression is associated with budding from a large group of cells from the cyst (Fig 5C and S3 Video), or the fragmentation of cysts into multiple smaller cell aggregates (Fig 5D, S4 Video). Staining for viability revealed that all cells were alive prior to detachment from cysts, and some cell survived outside of their cyst of origin (Fig 5H–5N). Expression of oncogenic KRAS (KRAS$^{G12V}$), which is an important driver of colon cancer [52], improved the survival of detached cells (Fig 5O–5S). Collectively, our results show that GIRDIN is required to maintain the cohesion of multicellular epithelial structures, and that its loss is associated with cell dispersion in control and cancer-mimetic contexts. This suggests that reduced GIRDIN levels may confer a more aggressive phenotype to cancer cells, and sustain tumor progression.

## Alteration of *GIRDIN* expression is associated with a poor prognosis in a subset of breast and lung cancers

To further investigate the relevance of our data to human cancer, we examined *GIRDIN* mRNA expression and patient survival outcomes. We observed that low *GIRDIN* expression was associated with reduced overall survival in more aggressive breast cancer types (Luminal

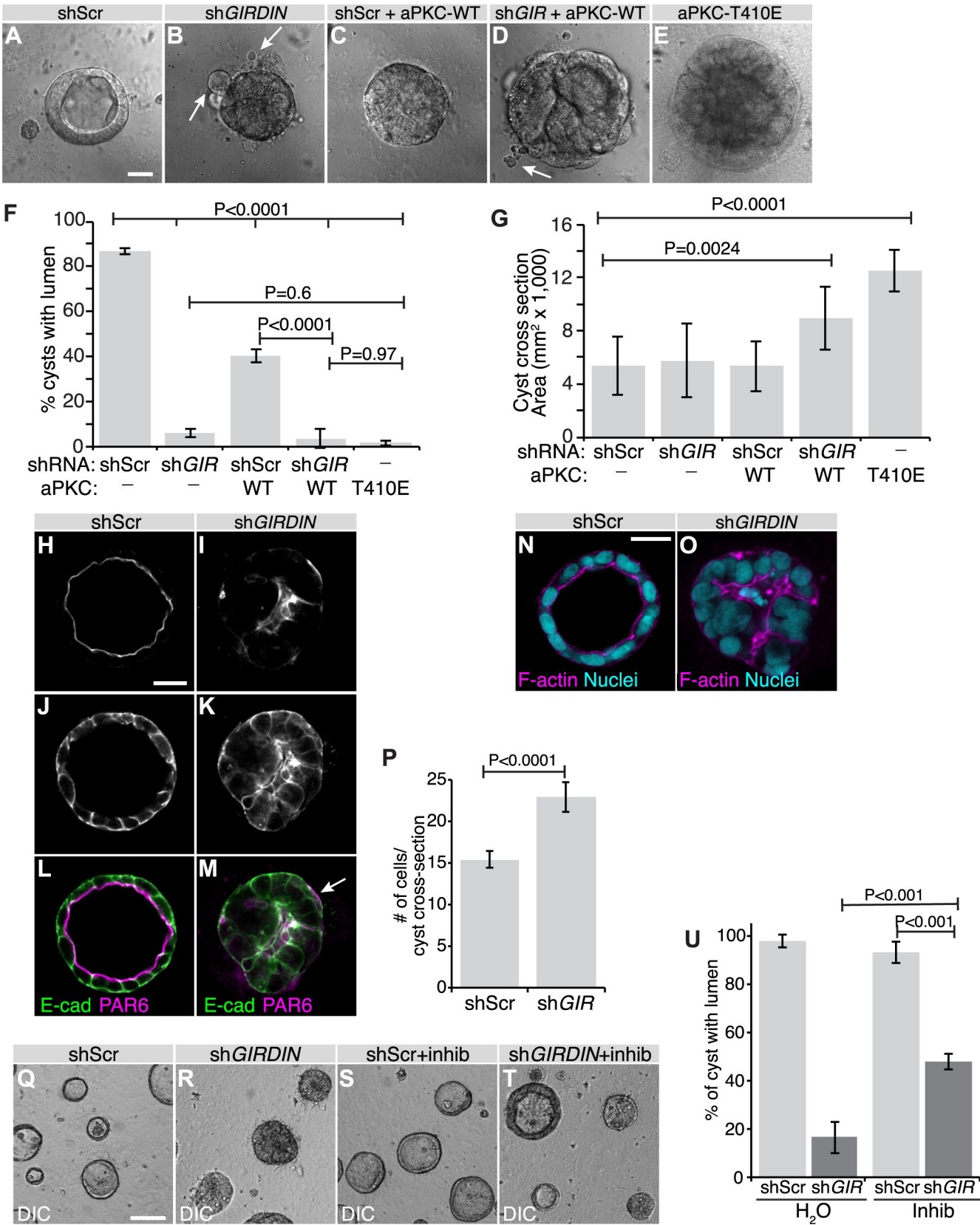

**Fig 4. Human GIRDIN is essential for epithelial morphogenesis and polarity. A-G,** Caco-2 cell cysts after 7 days in culture were observed by DIC microscopy, and the proportion of 3D cellular structures with a single prominent lumen and size were assessed (shScr, n = 27; sh*GIR*, n = 39, r = 3 independent experiments). Arrows in **B** and **D** highlight dissociated cells. Scale bar represents 25 μm. **F,** Histogram displaying the luminal phenotypes observed in A-E. Error bars = sd. **G,** Histogram displaying mean sizes (cross-sectional area) of cysts. Error bars = sd. **H-M,** Caco-2 cell cysts after 7 days in culture were immunostained for PAR6 and E-CAD and visualized by confocal microscopy. Arrows show an example of a structure with PAR6 localized basally. Scale bar represents 25 μm. **N-O,** Caco-2 cell cysts after 7 days in culture were stained with phalloidin (F-ACTIN) and Hoechst (nuclei) and visualized by confocal microscopy. Scale bar represents 25 μm. **P,** Histogram displaying the number of cells counted in cross-sectional images through the middle of 3D cellular aggregates. Error bars = sd. **Q-T,** Caco-2 cell cysts after 7 days in culture were visualized by DIC microscopy. Cells were treated with 1.5 μM of the aPKC inhibitor CRT-006-68-54 (inhib) or vehicle control (water) for the duration of the 3D culture period. Scale bar represents 100 μm. **U,** Histogram displaying the proportion of 3D cellular structures with a single prominent lumen (shScr, n = 278; sh*GIR*, n = 181; shScr+Inhib, n = 321; sh*GIR*+Inhib, n = 196; r = 3 independent experiments). Error bars = sd. Differences were determined using ANOVA with Tukey HSD (**F**, **G**, **U**) or Student's t-test (**P**).

B, HER2+, and basal Breast Cancer), compared to less aggressive Luminal A subtype (Fig 6A–6D). Low *GIRDIN* was also associated with poorer survival in lung adenocarcinoma, but not lung squamous cell carcinoma (Fig 6E and 6F). Therefore, *GIRDIN* expression is linked to survival outcomes, and displays specificity for certain cancer types.

## Discussion

Using classical genetics in flies, we have shown that mutation in *Girdin* exacerbates the polarity defects in zygotic *lgl* or *yrt* mutant embryos and concluded that Girdin is part of the lateral polarity network. We also found that Girdin opposes the function of aPKC, which plays a crucial role in the establishment and maintenance of the apical domain by antagonizing lateral proteins such as Lgl and Yrt [10,23]. We thus propose a model in which Girdin supports the activity of Yrt and Lgl by restricting the activity of aPKC (S4 Fig). Our work demonstrates that the role of Girdin in restricting aPKC activity is evolutionarily conserved. This function confers on human GIRDIN the ability to maintain apical-basal polarity in Caco-2 cells, and to support epithelial cyst morphogenesis. These results are in line with previous studies suggesting a role for GIRDIN in polarity and cystogenesis in MDCK and MCF10A epithelial cells [36,38]. It was shown that PKCλ enhances GIRDIN expression in MDCK cells [38]. Moreover, knockdown of *aPKC* or *GIRDIN* gives a similar phenotype characterized by defects in tight junction integrity and cyst formation [38,53–55]. It was thus proposed that GIRDIN is an effector of PKCλ [38]. Although cell-type-specific mechanisms may exist, our data suggest that this hypothesis needs to be revisited in favor of a model in which the induction of GIRDIN expression by PKCλ in MDCK cells initiates a negative feedback loop instead of cooperation between these proteins. The fact that both overactivation of aPKC or inhibition of its activity is deleterious to epithelial cell polarity and cyst morphogenesis may underlie the conflicting interpretations of the data in the literature [34,55]. GIRDIN is also known to modulate heterotrimeric G protein signaling [56,57]–a role that seems to contribute to the formation of normal cysts by MDCK cells [38]. In addition, it was demonstrated recently that GIRDIN acts as an effector of AMP-activated protein kinase (AMPK) under energetic stress to maintain tight junction function [39]. Of note, these two functions are not shared by fly Girdin [58,59], and were thus acquired by GIRDIN during evolution to fulfill specialized functions. In contrast, our discovery of the Girdin-dependent inhibition of aPKC reveals a core mechanism contributing to epithelial cell polarization from flies to humans.

GIRDIN is considered to be an interesting target in cancer due to its role in cell motility, and high levels of GIRDIN have been reported to correlate with a poor prognosis in some human cancers [60,61]. Notwithstanding that GIRDIN may favor tumor cell migration, our study indicates that inhibition of GIRDIN function in the context of cancer would be a double-edged sword for many reasons. Indeed, we showed that knockdown of *GIRDIN* exacerbates the impact of aPKC overexpression, and leads to overgrowth and lumen filling of Caco-2

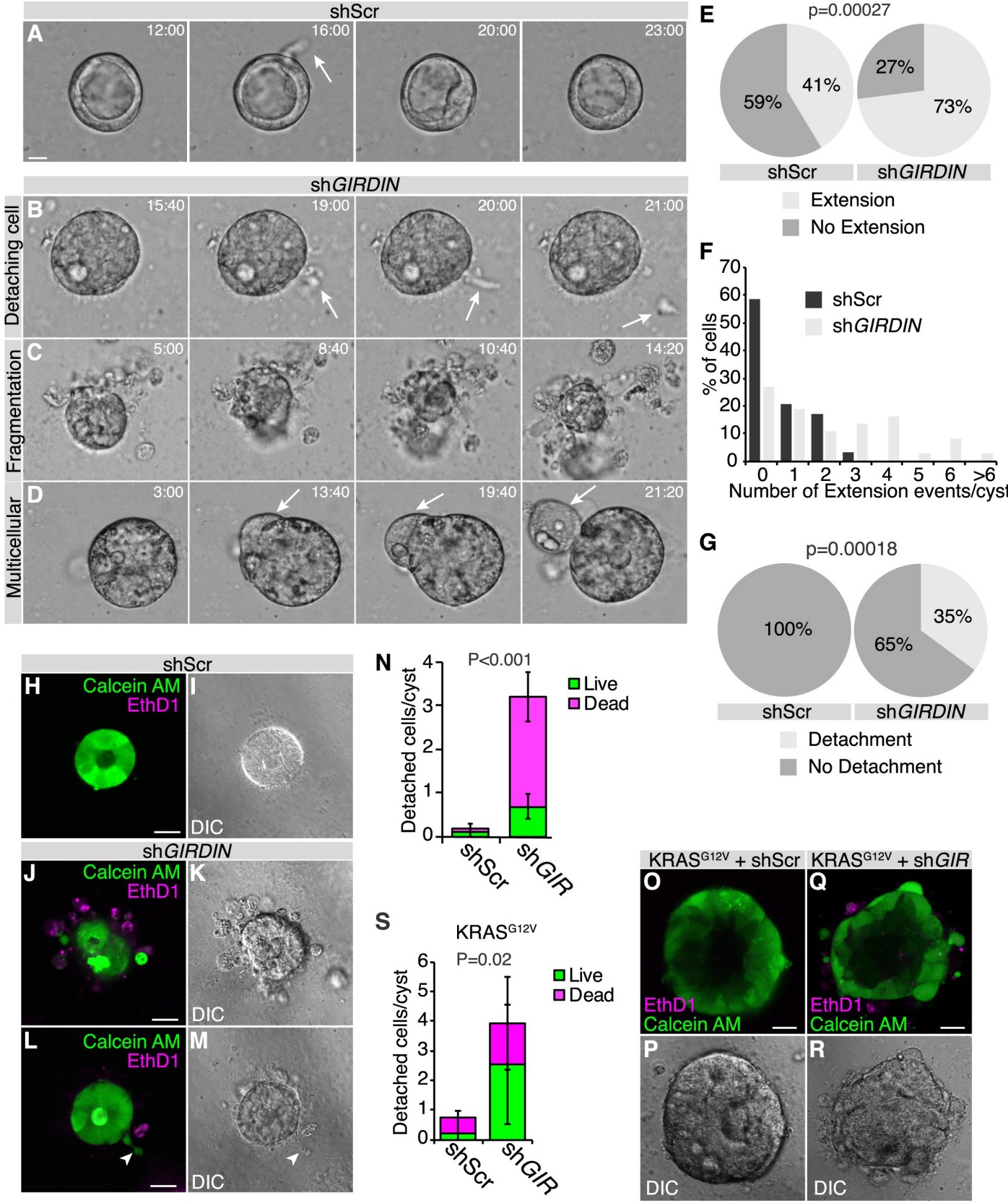

**Fig 5. Human GIRDIN prevents cell dissemination. A-D,** Caco-2 cells were cultured for 7 days, then observed by time-lapse confocal microscopy every 20 min for 26 h. Still frames from videos display representative events observed (shScr, n = 19; sh*GIR*, n = 28; r = 3 independent experiments). Scale bar represents 25 μm. **E,** Proportion of control (shScr) and *GIRDIN*-knockdown cysts displaying transient cellular extensions. **F,** Distribution of the number of transient cellular extensions per cyst. **G,** Proportion of control (shScr) and *GIRDIN*-knockdown cysts from which cells disseminated. **H-M,** *GIRDIN* knockdown and control (shScr) Caco-2 cells were cultured for 7 days, then stained with calcein AM and ethidium homodimer to visualize live (green) and dead (magenta) cells by confocal microscopy. Scale bars represent 25 μm. **N,** Histogram displaying the mean number of cells detached from Caco-2 cell aggregates (shScr, n = 27; sh*GIR*, n = 30; r = 2 independent experiments) and the proportion of live and dead detached cells. **O-R,** *GIRDIN* knockdown (n = 27, r = 2) and control (shScr, n = 27, r = 2) KRAS$^{G12V}$-expressing Caco-2 cells were stained with calcein AM and ethidium homodimer to visualize live (green) and dead (magenta) cells by confocal microscopy. Scale bars represent 25 μm. **S,** Histogram displaying the mean number of cells detached from cyst and the proportion of live and dead detached cells. Differences were determined using Chi-squared test (**E, G**) or ANOVA with Tukey HSD (**N, S**).

cell cysts. Of note, overexpression of aPKC can lead to cell transformation, and was associated with a poor outcome in several epithelial cancers [34,62]. Our study thus establishes that inhibiting GIRDIN in patients showing increased aPKC expression levels could worsen their prognosis. According to our data, abolishing GIRDIN function in tumor cells with decreased levels of the human Lgl protein LLGL1, as reported in many cancers [63], could also support the progression of the disease by altering the polarity phenotype. We observed cell detachment and dissemination from *GIRDIN* knockdown cysts, thus showing that GIRDIN is required for the cohesion of multicellular epithelial structures. Of note, cells, either individually or as clusters, detaching from cysts are alive and some of them remain viable. This is analogous to what was reported in *Girdin* mutant *Drosophila* embryos in which cell cysts detach from the ectoderm and survive outside of it [3]. Other phenotypes in *Girdin* mutant embryos are consistent with a role for Girdin in epithelial tissue cohesion, including rupture of the ventral midline and fragmentation of the dorsal trunk of the trachea. Mechanistically, Girdin strengthens cell–cell adhesion by promoting the association of core adherens junction components with the actin cytoskeleton [3]. A recent study established that this molecular function is evolutionarily conserved, and that GIRDIN favors the association of β-CATENIN with F-ACTIN [4]. Since knockdown of *GIRDIN* results in cell dispersion from Caco-2 cell cysts, and since weakening of E-CADHERIN-mediated cell–cell adhesion contributes to cancer cell dissemination and metastasis [64], it is plausible that reduced GIRDIN expression contribute to the formation of secondary tumors and cancer progression. This may explain why we found that low mRNA expression levels of *GIRDIN* correlates with decreased survival in more aggressive breast cancer subtypes and lung adenocarcinoma. Future studies using xenograft in mice, and investigating the expression of GIRDIN protein in cancer patients will help validating whether GIRDIN can repress the progression of certain types of epithelial cancers.

In conclusion, using a sophisticated experimental scheme combining *in vivo* approaches in *D. melanogaster* with 3D culture of human cells, we defined a conserved core mechanism of epithelial cell polarity regulation. Specifically, we showed that Girdin represses the activity of aPKC to support the function of Lgl and Yrt, and ensure stability of the lateral domain. This is of broad interest in cell biology, as proper epithelial cell polarization is crucial for the morphogenesis and physiology of most organs [10]. In addition, the maintenance of a polarized epithelial architecture is crucial to prevent various pathological conditions such as cancer progression [65]. Importantly, we show that normal GIRDIN function potentially impairs the progression of epithelial cancers by preserving cell polarity whilst restricting cell growth and cell dissemination. Thus, our results place a caveat on the idea that GIRDIN could be an interesting target to limit cancer cell migration, and indicate that inhibition of GIRDIN in the context of cancer could be precarious. Potential drugs targeting GIRDIN would thus be usable only in the context of precision medicine where a careful analysis of aPKC, LLGL1, and E-CAD expression, as well as the polarity status of tumor cells would be analyzed prior to treatment. Inhibition of GIRDIN in patients carrying tumors with altered expression of these proteins would likely worsen the prognosis.

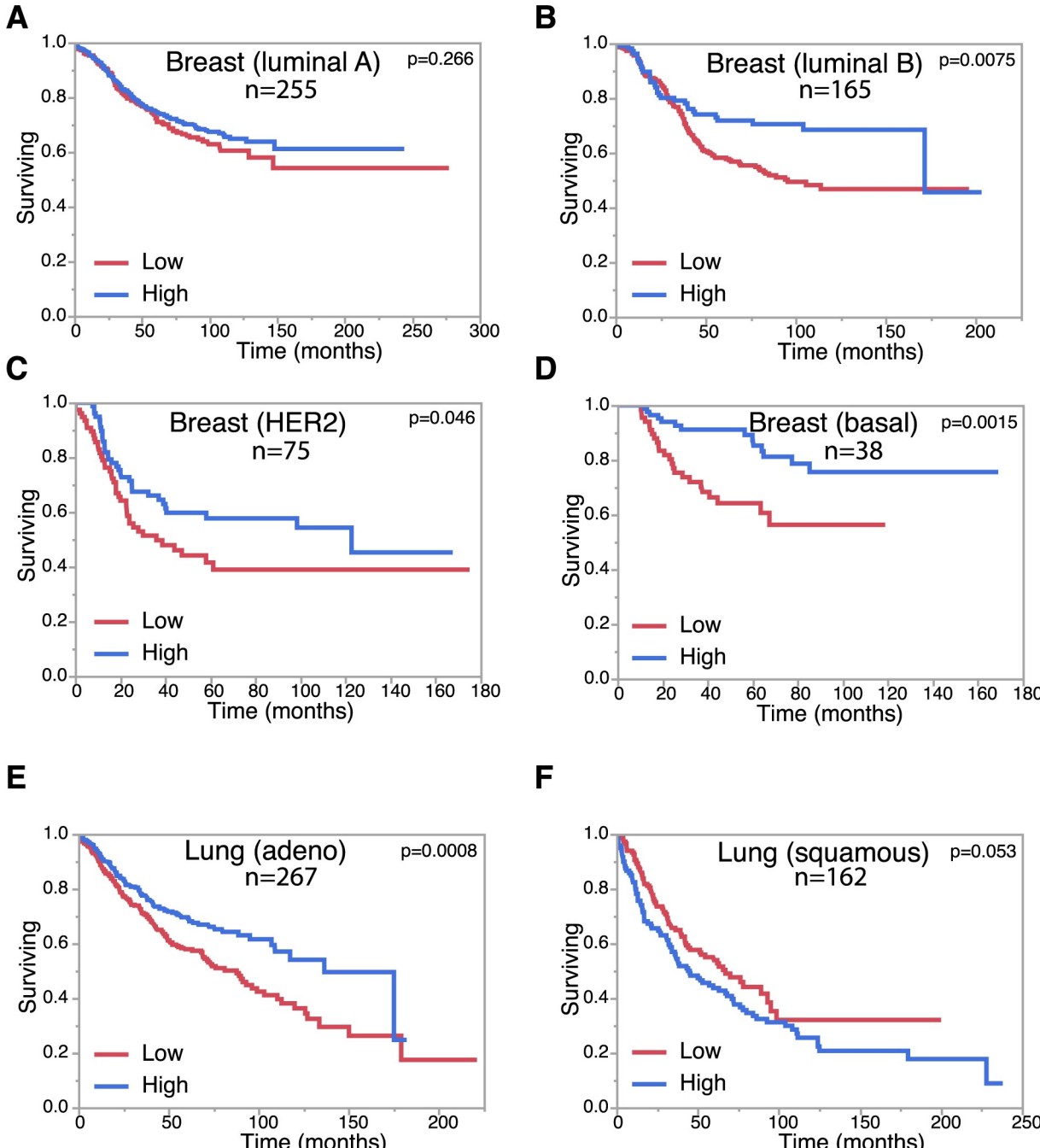

**Fig 6. Altered *GIRDIN* expression correlates with survival in epithelial cancers. A-F**, Kaplan-Meier survival plots for *GIRDIN* expression in breast cancer subtypes [luminal A (**A**), Luminal B (**B**), HER2-enriched (**C**), and basal (**D**)], and lung cancer subtypes [adenocarcinoma (**E**), squamous (**F**)]. p-values for differences between groups were determined using a Log-Rank test.

## Materials and methods

### *Drosophila* genetics

*The following mutant alleles and transgenic lines were used in this study: Girdin$^2$ [3], lgl$^4$ [66], yrt$^{75}$ [27], UAS-GFP.nls (Bloomington Drosophila Stock Center [BDSC] Stock No. 4776), and*

UAS-FLAG-Girdin [3]. Germ line clone females were produced using the FLP-DFS technique [67], and were used to produce maternal and zygotic (M/Z) mutant embryos. Maternal knock-down of *aPKC* and *lgl* was achieved by crossing the driver line matαtub67;15 (obtained from D. St-Johnston, University of Cambridge, Cambridge, UK) to a line containing an inducible shRNA directed against *aPKC* (BDSC Stock No. 34332) or *lgl* (BDSC Stock No. 35773) at 25˚C.

## 3D cell culture

Human intestinal epithelial cell line Caco-2 cells were purchased from American Type Culture Collection (ATCC). Caco-2 cells were cultured at 37˚C in 5% $CO_2$ in DMEM (Wisent) supplemented with 10% fetal bovine serum (Wisent), 100 U/ml penicillin (Wisent), 0.1 mg/ml streptomycin (Wisent), and 2 mM L-glutamine (Wisent). For 3D culture, cells were seeded onto Geltrex-coated dishes at a density of $1.25 \times 10^4$ cells per well and were maintained in 2% Geltrex in complete medium at 37˚C under humidified atmosphere of 5% $CO_2$. After 10 days in adherent culture, cells were collected for further experiments. For inhibitor studies 1.5 μM CRT-006-68-54 or water control was added to culture medium at the time of seeding cells in 3D cultures. All cell lines were routinely verified to be mycoplasma free.

## Lentivirus production and shRNA

Human embryonic kidney cell line HEK293LT (ATCC) were cultured at 37˚C in 5% $CO_2$ in DMEM supplemented with 10% fetal bovine serum, 100 U/ml penicillin, 0.1 mg/ml streptomycin. Lentiviruses were produced by calcium phosphate transfecting HEK293LT cells in 10-cm dishes with 20 μg of lentiviral plasmid, 15 μg of packaging plasmid (psPAX2; Addgene plasmid 12260), and 6 μg of VSVG coat protein plasmid (pMD2.G; Addgene plasmid 12259; both from D. Trono, Ecole Polytechnique Federale de Lausanne, Switzerland). Viral supernatants were collected after 48 h, aliquoted, and frozen at –80˚C. shRNAs targeting the human *GIRDIN* mRNA were obtained in pLKO.1-puro vectors from the RNAi Consortium (TRC). Caco-2 cells were infected with lentiviral supernatants and selected by the addition of 20 μg/ml puromycin. The target sequences of TRC clone number TRCN0000148551 is CCGGCTTCAT TAGTTCTGCGGGAAACTCGAGTTTCCCGCAGAAC TAATGAAGTTTTTTG.

## Immunofluorescence on *Drosophila* embryos

Embryos were dechorionated in 3% sodium hypochlorite for 5 min, rinsed in water, and submitted to a flash heat fixation by sequential addition of 5 ml of E-wash buffer (7% NaCl, 0.5% Triton X-100) at 80˚C and 15 ml of E-wash buffer at 4˚C. Embryos were then washed with PBS prior to devitellinization by strong agitation in methanol under a heptane phase (1:1), and further incubated in fresh methanol for 1 h. Saturation of non-specific binding sites was achieved by a 1-h incubation in NGT (2% normal goat serum, 0.3% Triton X-100 in PBS), which was also used to dilute primary antibodies. The following primary antibodies were used overnight at 4˚C, under agitation: rat anti-Crb [68], 1:250; mouse anti-Dlg1 [clone 4F3, Developmental Studies Hybridoma Bank (DSHB)], 1:25; mouse anti-FLAG (M2, Sigma), 1/250; rabbit anti-Lgl D300 (Santa Cruz Biotechnology) 1:100. Embryos were washed three times for 20 min in PBT (0.3% Triton X-100 in PBS) before and after incubation with secondary antibodies (1:400 in NGT, 1 h at room temperature), which were conjugated to Cy3 (Jackson ImmunoResearch Laboratories) or Alexa Fluor 488 (Molecular Probes). Embryos were mounted in Vectashield mounting medium (Vector Labs), and imaged with a FV1000 confocal microscope coupled to FluoView 3.0 (Olympus), using a 40× Apochromat lens with a numerical aperture of 0.90. Images were uniformly processed with Olympus FV1000 viewer (v.4.2b), ImageJ (National Institutes of Health), or Photoshop (CC 2017; Adobe).

## Cuticle preparation

Embryos were dechorionated, mounted in 100 μl of Hoyer's mounting medium (prepared by mixing 50 ml of distilled water, 20 ml of glycerol, 30 g of gum Arabic, and 200 g of chloral hydrate)/lactic acid (1:1) and incubated overnight at 80˚C. Embryos were imaged with an Eclipse 600 microscope (Nikon) through a 10× Plan Fluor objective with a numerical aperture of 0.30, and a CoolSNAP fx camera (Photometrics) coupled to MetaVue 7.77 (Molecular Devices). Images were processed with Photoshop (CC 2017; Adobe).

## Antibody production

Antibodies against the amino acids 696 to 972 of Yrt in fusion with GST were produced in rabbits (Medimabs).

## Phosphatase assays

Dechorionated embryos were homogenized in ice-cold lysis buffer (1% Triton X-100, 50 mM TRIS-HCl pH 7.5, 5% glycerol, 150 mM NaCl, 1 mM PMSF, 0.5 μg/mL aprotinin, 0.7 μg/mL pepstatin, and 0.5 μg/mL leupeptin). Lysates were cleared by centrifugation at 4˚C, and 400 units of λ Phosphatase (New England Biolabs) was added to 50 μg of proteins extracted from embryos. The volume of the reaction mix was completed to 30 μl with the MetalloPhosphatase buffer (New England Biolabs) containing 1 mM of $MnCl_2$ prior to a 30-min incubation at 30˚C. The reaction was stopped by addition of Laemmli's buffer.

## Western blot

Dechorionated embryos were homogenized in ice-cold lysis buffer (1% Triton X-100, 50 mM TRIS HCl pH 7.5, 5% glycerol, 100 mM NaCl, 50 mM NaF, 5 mM EDTA pH 8, 40 mM β-glycerophosphate, 1 mM PMSF, 0.5 μg/mL aprotinin, 0.7 μg/mL pepstatin, 0.5 μg/mL leupeptin and 0.1 mM orthovanadate) and processed for SDS-PAGE and western blotting as previously described (Laprise et al., 2002). Primary antibodies used were: guinea pig anti-Girdin 163 [3], 1:2,000; rabbit anti-Yrt (this study), 1:5000; mouse anti-Gapdh1 (Medimabs), 1:500; rabbit anti-Lgl D300 (Santa Cruz Biotechnology) 1:1,000; rabbit anti-PKCζ C20 (Santa Cruz Biotechnology), 1:2,000; mouse anti-Actin (Novus Biologicals), 1:10,00; mouse anti- β−Tubulin (DM1A, Sigma), 1:10,000; rabbit anti-GIRDIN (ABT80, Millipore), 1:1,000. HRP-conjugated secondary antibodies were used at a 1:2,000 to 1:10,000 dilution.

## Chemical treatment of embryos

Dechorionated embryos were incubated with 500 μM of the aPKC inhibitor CRT-006-68-54 (diluted in PBS with 0,9% NaCl) under an octane phase (1:1) for 1h at room temperature.

## Immunofluorescence on human 3D cysts

Three-dimensional cysts were transfected with plasmids and fixed with 2% paraformaldehyde/ PBS for 10 min and permeabilized in PBS supplemented with 0.5% Triton X-100/10% goat serum for 1 h, and incubated overnight in primary antibodies. The following primary antibodies were used: mouse anti-PAR6 (1:100, Santa Cruz Biotechnology); rabbit anti-E-cadherin (1:100, Cell Signaling Technology). Cysts were washed three times for 15 min in 0.5% Triton X-100/PBS before and after incubation with secondary antibodies. Proteins were visualized with 647 Phalloidin (1:100, Invitrogen), Cy3-Donkey anti-Rabbit (1:750, Jackson IR through Cedarlane), Alexa Fluor 647-Donkey anti-Rabbit (1:200, Jackson IR through Cedarlane). DNA was detected with Hoechst dye 33258 (Sigma Aldrich). Cysts were imaged with ZEISS LSM700

confocal microscope at Plan-Apochromat 20X/0.8 M27 objective lens. Images were uniformly processed with ImageJ (National Institutes of Health).

## Cell survival assays

CaAM (calcein AM)/EthD-I (ethidium homodimer I) staining was performed in three-dimensional cysts with 2 mM calcein AM and 4 mM ethidium homodimer I (Live/Dead Viability/Cytotoxicity kit; Life Technologies) for 40 min at 37°C in the dark. DNA was detected with Hoechst dye 33312 (Invitrogen).

## Live imaging

Three-dimensional cysts were imaged using ZEISS LSM700 confocal microscope at 20X/0.4 Korr M27 objective lens. Cells were imaged every 20 min for 26 h in a humidified chamber with 5% $CO_2$ and heated to 37°C.

## Patient cancer survival

Survival data was retrieved from the kmplot resource (kmplot.com; [69]) for breast, lung, ovarian, and gastric mRNA expression. Jetset optimal probes were selected for analysis. The breast cancer intrinsic subtypes were selected for Luminal A, Luminal B, HER2-positive, and basal cancers. Lung cancer subtypes were selected using the histology selection option for adenocarcinoma and squamous lung cancers. Survival plots with were generated using JMP14 statistical software.

## Statistical analysis

Pairwise statistics were performed using Student's tests. Multiple comparisons were compared using ANOVA, with Tukey HSD tests. Distributions were examined using a chi-square goodness of fit test. Differences in survival were determined using a Log-Rank test. Statistical analyses were performed with JMP14 statistical software, with $\alpha = 0.05$ for all tests.

## Supporting information

**S1 Fig. Weakening of the *zonula adherens* is not sufficient to cause polarity defects in *lgl* mutant embryos. A-D**, Cuticle preparations of whole mounted embryos of the indicated genotypes. Scale bar in **A** represents 100 μm, and also applies to **B-D**. **E-P**, Embryos at embryonic stage (St) 11, 13 or 16 were fixed and processed for immunofluorescence. The distribution of the apical marker Crb and of the lateral protein Dlg1 was then assessed by confocal microscopy in the ventral ectoderm or the ventral epidermis. Scale bar in **E** represents 10 μm, and also applies to **F-P**. A-P show representative results of experiments that were performed in triplicate. At least 20 embryos were analyzed in each replicate.
(TIF)

**S2 Fig. Related to Fig 3. A-B**, Western blots showing knockdown efficiency for aPKC (A), and GIRDIN (B). Actin (A) or TUBULIN (B) were used as loading control. **C**, Embryos expressing FLAG-Girdin were fixed and immunostained with anti-FLAG antibodies. **D-E**, Crb (green) and Lgl (magenta) distribution in a wild type embryo (D) or a FLAG-Girdin expressing specimen (E). Panels depict whole embryo view (anterior is to the left, and dorsal is up). Scale bar in C = 10 μm, scale bar in D, E = 100 μm.
(TIF)

**S3 Fig. Related to Fig 4. Kinase-deficient aPKC restores lumen formation in GIRDIN-deficient cells. A-F**, Caco-2 cell cysts after 7-days in culture were visualized by DIC microscopy. GIRDIN-deficient (sh*GIR*) or control cells (shScr) expressed GFP (control), wild-type (aPKC-WT), or kinase-deficient (aPKC-K281W) aPKC. Scale bars represent 50 μm. **G**, Histogram displaying the proportion of 3D cellular structures with a single prominent lumen (shScr/GFP, n = 486; shScr/aPKC-K281W, n = 309; shScr/aPKCWT, n = 341; sh*GIR*/GFP, n = 292; sh*GIR*/aPKC-K281W, n = 229; sh*GIR*/aPKC-WT, n = 227; r = 3 independent experiments). Error bars = sd. Differences were determined using ANOVA with Tukey HSD. (TIF)

**S4 Fig. Girdin restricts aPKC activity to maintain apical-basal polarity.** Our data indicate that Girdin cooperates with Lgl and Yrt. The latter two proteins act in parallel pathways to antagonize the apical machinery, thereby supporting lateral membrane stability [8,27,28,30]. Although Lgl and Yrt act independently, they have in common that they are both negatively regulated by aPKC [15,24,25]. We provide evidence that Girdin antagonizes aPKC function. We thus propose a model in which Girdin supports the function of Yrt and Lgl by restricting the activity of aPKC. (TIF)

**S1 Video. Imaging of a Caco-2 cell cyst.** Live imaging of a wild type Caco-2 cell cyst, which was imaged every 20 min for 26 h. (AVI)

**S2 Video. Cells detach from *GIRDIN* knockdown epithelial cysts.** A *GIRDIN* knockdown Caco-2 cell cyst was imaged every 20 min for 26 h. (AVI)

**S3 Video. Cell clusters are extruded from *GIRDIN* knockdown cell cysts.** A *GIRDIN* knockdown Caco-2 cell cyst was imaged every 20 min for 26 h. (AVI)

**S4 Video. GIRDIN maintains the cohesion of epithelial structures.** Live imaging of a *GIRDIN* knockdown Caco-2 cell cyst. The latter was imaged every 20 min for 26 h. (AVI)

## Acknowledgments

The authors would like to acknowledge D. Bilder, U. Tepass, D. St-Johnston, the Bloomington *Drosophila* Stock Center, the *Drosophila* Genomics Resource Center, and the Developmental Studies Hybridoma Bank for reagents. Flybase was used as an important database for this work.

## Author Contributions

**Conceptualization:** Li-Ting Wang, Myriam Sévigny, Elise Houssin, Luke McCaffrey, Patrick Laprise.

**Data curation:** Cornélia Biehler, Li-Ting Wang, Myriam Sévigny, Alexandra Jetté, Clémence L. Gamblin, Rachel Catterall, Elise Houssin, Patrick Laprise.

**Formal analysis:** Cornélia Biehler, Li-Ting Wang, Myriam Sévigny, Alexandra Jetté, Clémence L. Gamblin, Rachel Catterall, Elise Houssin, Luke McCaffrey.

**Funding acquisition:** Luke McCaffrey, Patrick Laprise.

**Investigation:** Cornélia Biehler, Li-Ting Wang, Myriam Sévigny, Alexandra Jetté, Clémence L. Gamblin, Patrick Laprise.

**Methodology:** Cornélia Biehler, Li-Ting Wang, Patrick Laprise.

**Project administration:** Luke McCaffrey, Patrick Laprise.

**Resources:** Luke McCaffrey, Patrick Laprise.

**Supervision:** Luke McCaffrey, Patrick Laprise.

**Writing – original draft:** Patrick Laprise.

**Writing – review & editing:** Cornélia Biehler, Myriam Sévigny, Alexandra Jetté, Clémence L. Gamblin, Elise Houssin, Luke McCaffrey, Patrick Laprise.

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
