## [Decision Letter · Decision Letter 0]

2 Oct 2019

Dear Patrick,

Thank you very much for submitting your Research Article entitled 'Girdin is a component of the lateral polarity protein network restricting cell dissemination' to PLOS Genetics. Your manuscript was fully evaluated at the editorial level and by three independent peer reviewers. The reviewers appreciated the attention to an important problem, but raised some substantial concerns about the current manuscript. Based on the reviews, we will not be able to accept this version of the manuscript, but we would be willing to review again a much-revised version. We cannot, of course, promise publication at that time.

Should you decide to revise the manuscript for further consideration here, your revisions should address in some way each of the specific points made by each reviewer. From our analysis of the reviews, there are two major issues that need to be addressed, each involving strengthening one of the major conclusions of the manuscript.  Reviewers 1 and 2 point out that the evidence using epistasis for this being a simple linear pathway is weak . Reviewer 2 offers clear suggestions for how to strengthen this point.  Second, both Reviewers 2 and 3 feel that the argument that Girdin represses aPKC activity is weak, and both request some more direct evidence that this is the case.  Reviewer 2 requests some additional quantification, which should be straightforward.  We feel that Reviewer 3's requests for more analysis of the cancer data go beyond the scope of the current manuscript, and should be addressed by adjusting the text of that section.  Having addressed these issues, you will need to provide a detailed list of your responses to the review comments and a description of the changes you have made in the manuscript.

If you decide to revise the manuscript for further consideration at PLOS Genetics, please aim to resubmit within the next 60 days, unless it will take extra time to address the concerns of the reviewers, in which case we would appreciate an expected resubmission date by email to plosgenetics@plos.org.

Accompanying reviewer attachments are included with this email; please notify the journal office if any appear to be missing. They will also be available for download from the link below. You can use this link to log into the system when you are ready to submit a revised version, having first consulted our Submission Checklist.

[LINK]

We are sorry that we cannot be more positive about your manuscript at this stage. Please do not hesitate to contact us if you have any concerns or questions.

Yours sincerely,

Mark Peifer

Guest Editor

PLOS Genetics

Gregory P. Copenhaver

Editor-in-Chief

PLOS Genetics

Reviewer's Responses to Questions

**Comments to the Authors:**

Reviewer #1: Biehler et al.

This is an interesting manuscript. The authors identify a genetic requirement for Girdin protein in epithelial cell polarity in Drosophila and mammalian cells. Girdin acts in parallel or upstream of Lgl and Yurt to maintain basolateral identity and repress spreading of the apical determinants in Drosophila. In polarised epithelial cysts in culture, RNAi knockdown of GIRDIN also disrupts normal cyst polarisation, causing collapse of lumens and fragmentation. Finally, they implicate GIRDIN levels in various cancers.

Overall, these are some nice phenotypic observations that deserve publication in PLoS Genetics after addressing some points.

Minor comments:

1. In the results section, the authors state that Girdin is epistatic to Yurt, which acts downstream of Girdin. It is quite hard to establish robust epistasis in polarity, as it normally requires expression of the downstream component to produce a phenotype in the mutant background of the upstream component. Furthermore, epistasis is hard to do properly in the presence of maternal Girdin protein. Nevertheless, I understand what they are trying to say, and would just suggest modification of the text to soften the interpretation. If the authors want to conclude that Girdin is upstream of Lgl and Yurt, they would need to show mis-localisation of Lgl and Yurt in Girdin mutants, and also examine Girdin localisation in Lgl and Yurt mutants.

2. What's the effect of expressing Flag-Girdin alone in Fig 4? Presumably no effect, so it should be possible to examine the Flag-Girdin subcellular localisation in these embryos.

3. In the introduction, the authors mention many key apical determinants in Drosophila epithelia except for Cdc42. Any reason why this is left out?

Reviewer #2: Biehler et al describe genetic interactions between Girdin and cell polarity regulators in both Drosophila embryos and 3D cultures of Caco-2 cells. They also show effects of Girdin depletion on the structure of Caco-2 cell cysts, and describe a correlation between low Girdin expression and poor prognoses for certain cancer types. In Drosophila embryos, for girdin and lgl mutants, or girdin and yurt mutants, certain double mutant phenotypes are shown to be worse than either corresponding single mutant phenotype. In contrast, girdin loss-of-function is shown to suppress the phenotype of aPKC RNAi embryos, and also to reduce phosphorylated species of Yurt. In Caco-2 cell culture, girdin RNAi samples are shown to share defects of aPKC over-expression samples, and the combination of girdin RNAi plus aPKC over-expression mimics the even worse effects of over-expressing constitutive active aPKC (abnormal cyst structure and increased cyst size). The cyst defects of girdin RNAi are also shown to be suppressed with application of an aPKC inhibitor. The girdin RNAi cysts are shown to shed cells whose survival can be enhanced with additional expression of KRASG12V. Finally, the authors derive a correlation between low girdin expression and poor prognoses for certain cancers by analyzing publicly available data. The study is of potential interest to those studying epithelial cell biology and cancers, but substantial issues exist.

1. Alternate interpretations are possible for the Drosophila genetic interaction studies. The following conclusions of the authors are in question:

1a. “Thus, these genes act in a common genetic pathway in which yrt acts downstream of Girdin”; and “Our results thus suggest that lgl is epistatic to Girdin, and that lgl is downstream of Girdin in this newly defined pathway.”

In these cases, lack of enhancement of the yrt null allele m/z phenotype, or the lgl maternal RNAi phenotype, by zygotic null mutants of girdin is taken as evidence for a single pathway, and of girdin being upstream. However, the girdin reduction is not null because of the maternal supplies of the normal gene product. It remains possible that the phenotype could be worsened in a fully null context for both yrt and girdin, or for both lgl and girdin. Also, clear conclusions about upstream-downstream relationships cannot be made from these data (combining gain-of-function and loss-of-function effects could allow clearer conclusions, as could incorporation of protein localization studies with the mutant analyses).

Other data from the authors indicate that Girdin does not sit atop a single pathway upstream of Yrt or Lgl. For one, the phenotype of girdin m/z mutants is much milder than that of either lgl m/z mutants or yrt m/z mutants. Also, the relationship between Yurt and Lgl is unclear—the yurt null m/z phenotype is weaker than the lgl knock-down phenotype. How could Girdin be restricted to a single pathway with either Yrt or Lgl if these players have different loss-of-function phenotypes and Girdin genetically interacts with both of them?

It seems all the authors can conclude is that Girdin cooperates positively with Yrt and Lgl for epithelial cell polarity and development. Bases for the cooperation remain unclear, and could be through previously reported roles for Girdin (e.g. effects on AJs). The authors say (with data not shown) that they “observed no genetic interaction between genes coding for core components of the ZA and lgl”, but lack of a genetic interaction is not necessarily conclusive, and thus these ZA data should be strengthened with positive controls and shown.

1b. “we showed that Girdin represses the activity of aPKC to support the function of Lgl and Yrt”; and “Girdin-dependent inhibition of aPKC reveals a core mechanism contributing to epithelial cell polarization”.

The authors’ data indicate that Girdin has effects that are antagonistic to those of aPKC, but the mechanism(s) involved remain unclear. An alternate possibility is that Girdin could antagonize the effects of aPKC by supporting the effects of Lgl and Yrt (upstream-downstream relationships are not yet clear). Also, no evidence is provided for Girdin inhibiting the kinase activity of aPKC (phosphorylated species of Yrt are shown to be increased in girdin mutant embryo samples but it is unclear if aPKC phosphorylation is generating these species).

2. The number of experimental replicates, and embryo numbers assessed in each replicate, need to be provided to show the penetrance of effects in the in situ staining studies and cuticle analyses in Figures 1-3.

3. For Figure 4E-F, quantifications of the degree of embryo lethality should accompany the quantifications of the cuticle phenotypes of the embryos that have died. Are the genetic interactions also affecting the degree of induced lethality? This applies to Figures 1-3 as well.

4. For the Caco-2 cyst experiments, the number of separate experiments conducted should be indicated in all cases.

5. The genotype labelled for Fig 5C is not consistent with that in Fig 5F and G ("shScr” is missing in 5C).

6. For the experiment with the aPKC inhibitor, specificity should be confirmed with aPKC RNAi.

7. N values, and thus the statistical tests, are unclear for Figure 7. More explanation is needed.

Reviewer #3: Biehler et al report roles of Girdin in epithelial polarity and cell-cell adhesion, combining in vivo genetic approaches in D. melanogaster with 3D culture of human cancer cells and clinical data set. Girdin is known to regulate several cellular processes such as actin dynamics, cell motility, heterotrimeric G protein activity, postnatal angiogenesis and neurogenesis. In this study, the authors showed that Girdin plays a crucial role in lateral polarity by inhibiting aPKC. Using Drosophila mutants, they conclude that Girdin is an upstream regulator of lateral polarity proteins, Yrt and Lgl. Also, loss of Girdin expression resulted in aPKC-dependent overgrowth of tumor-like structure (disorganized cell cysts) in 3D culture of human intestinal epithelial Caco-2 cells, supporting Girdin’s function in cell polarity. Moreover, lack of Girdin reduced cell-cell cohesion and increased the dissemination of individual cells or cell clusters from Caco-2 cell cysts and Kras-V12-induced tumorspheres, suggesting the possibility that Girdin could restrict cancer progression and metastasis. Finally, authors argue that reduced Girdin levels correlate with a poor outcome in patients with aggressive breast and lung cancers.

The study on Girdin’s function in epithelial polarity using Drosophila mutants seems to be logically sound. However, authors’ claim that reduced Girdin’s level correlates with poor outcome in cancer patients is not supported from the presented data. More experiments are needed to address this point.

Major comments

1． Authors showed that lack of Girdin expression enhanced the dissemination of individual cells or cell cluster from Caco-2 cells and Kras V12-induced tumorspheres. This finding does not necessarily mean that lack of Girdin promotes cancer metastasis. They argue that this correlates with poor outcome of certain cancer patients without any experimental evidence. There is a big gap between cell biological finding and clinical data. To address this point, authors should perform further in vitro and in vivo experiments including mouse xenograft models and 3D-invasion assay of cancer cells using matrigel.

2． In Fig. 7, analysis of patient survival outcome is based on mRNA levels of Girdin. The correlation between Girdin and patient outcome must be validated by investigating Girdin proteins expression in cancer cells with immunohistochemical analysis.

3． How does Girdin regulate aPKC activity? Is there direct association between two molecules? In addition to cell biological analysis shown in Fig. 5, biochemical analysis is needed to confirm direct or indirect regulation of aPKC activity by Girdin.

**Have all data underlying the figures and results presented in the manuscript been provided?**

Reviewer #1: Yes

Reviewer #2: None

Reviewer #3: Yes

PLOS authors have the option to publish the peer review history of their article (what does this mean?). If published, this will include your full peer review and any attached files.

Reviewer #1: No

Reviewer #2: No

Reviewer #3: No

---

## [Decision Letter · Decision Letter 1]

14 Feb 2020

Dear Patrick

We are pleased to inform you that your manuscript entitled "Girdin is a component of the lateral polarity protein network restricting cell dissemination" has been editorially accepted for publication in PLOS Genetics. Congratulations!

Yours sincerely,

Mark Peifer

Guest Editor

PLOS Genetics

Gregory P. Copenhaver

Editor-in-Chief

PLOS Genetics

Comments from the reviewers (if applicable):

Reviewer's Responses to Questions

Comments to the Authors:

Please note here if the review is uploaded as an attachment.

Reviewer #2: My past comments have been addressed effectively. This is an important paper that should be of substantial interest.

Have all data underlying the figures and results presented in the manuscript been provided?

Large-scale datasets should be made available via a public repository as described in the 

PLOS Genetics

data availability policy, and numerical data that underlies graphs or summary statistics should be provided in spreadsheet form as supporting information.

Reviewer #2: None

PLOS authors have the option to publish the peer review history of their article (what does this mean?). If published, this will include your full peer review and any attached files.

Do you want your identity to be public for this peer review?

 For information about this choice, including consent withdrawal, please see our Privacy Policy.

Reviewer #2: No

**Data Deposition**

http://datadryad.org/submit?journalID=pgenetics&manu=PGENETICS-D-19-01511R1

Press Queries

---

## [Editor Report · Acceptance letter]

13 Mar 2020

PGENETICS-D-19-01511R1 

Girdin is a component of the lateral polarity protein network restricting cell dissemination 

Dear Dr Laprise, 

We are pleased to inform you that your manuscript entitled "Girdin is a component of the lateral polarity protein network restricting cell dissemination" has been formally accepted for publication in PLOS Genetics! Your manuscript is now with our production department and you will be notified of the publication date in due course.

With kind regards,

Jason Norris

PLOS Genetics

On behalf of:
